# AMDP: Asynchronous Multi-Directional Pipeline Parallelism for Large-Scale Models Training

**Ling Chen** [1 2]   **Houming Wu** [1 2]   **Wenjie Yu** [1 2]

## Abstract

Pipeline parallelism is essential for large-scale model training, but existing asynchronous approaches often degrade convergence due to parameter mismatch between forward and backward passes. We propose **A**synchronous **M**ulti-**D**irectional **P**ipeline parallelism (**AMDP**) to mitigate this issue while sustaining high utilization. AMDP limits the first stage of each pipeline to process at most two minibatches before backpropagation, bounding the number of parameter updates between forward and backward passes. To alleviate the resulting pipeline bubbles, AMDP launches multiple concurrent pipelines and adapts their number according to pipeline depth. In addition, AMDP accumulates gradients across minibatches and applies them in a single update, ensuring that only a bounded number of minibatches experience parameter mismatch, limited to within one optimization step. Experiments on GPT- and BERT-style models demonstrate that AMDP significantly accelerates training while preserving convergence.

## 1. Introduction

Since the introduction of Transformers (Vaswani et al., 2017), scaling model size has become a primary driver of progress in deep learning, enabling remarkable advances across diverse applications. Larger models consistently deliver stronger performance, but their training poses formidable challenges. For example, GPT-3 (Brown et al., 2020) contains 175B parameters, requiring about 350 GB of memory in 16-bit precision. Even disregarding memory constraints, training GPT-3 on a single NVIDIA V100 GPU

[1]State Key Laboratory of Blockchain and Data Security, Zhejiang University, Hangzhou, China [2]College of Computer Science and Technology, Zhejiang University, Hangzhou, China. Correspondence to: Ling Chen <lingchen@cs.zju.edu.cn>.

*Proceedings of the 43rd International Conference on Machine Learning*, Seoul, South Korea. PMLR 306, 2026. Copyright 2026 by the author(s).

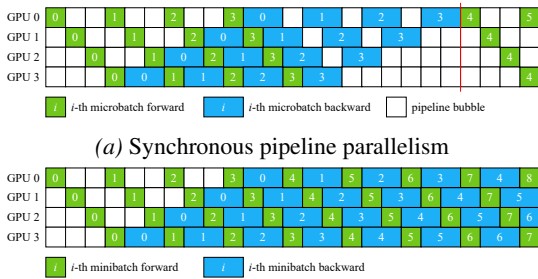

*(a)* Synchronous pipeline parallelism

*(b)* Asynchronous pipeline parallelism

*Figure 1.* Comparison of synchronous and asynchronous pipeline parallelism, with four model stages assigned to four devices. The backward pass duration is assumed to be twice that of the forward pass.

would take an estimated 288 years (Narayanan et al., 2021b). These prohibitive costs necessitate distributed parallel training across multi-GPU clusters.

Pipeline parallelism (Guan et al., 2024) has emerged as a key strategy for distributed parallel training. By partitioning a model into sequential stages, each executed on a different GPU, pipeline parallelism reduces per-device memory requirements and increases throughput. Depending on how parameters are updated, pipeline parallelism can be classified into synchronous and asynchronous schemes. In synchronous pipeline parallelism (Huang et al., 2019), a minibatch is split into microbatches. Gradients from all microbatches are accumulated and applied only after all backward passes complete, and the next iteration begins once all parameters are updated. As shown in Figure 1a, this scheme is straightforward but leads to pipeline bubbles—idle periods caused by inter-stage dependencies.

Asynchronous pipeline parallelism (Narayanan et al., 2019) continuously feeds minibatches into the pipeline, updating parameters immediately after each backward pass. As shown in Figure 1b, this design eliminates bubbles once steady state is reached, but sacrifices convergence because parameter updates may occur between the forward and backward passes of a minibatch, resulting in parameter mismatch. To mitigate this mismatch, two techniques have been introduced: parameter stash and parameter prediction. Parameter stash (Narayanan et al., 2019) preserves the parameters used

in the forward pass and reuses them in the corresponding backward pass to enforce consistency. Parameter prediction (Chen et al., 2018; Ajanthan et al., 2025) instead forecasts future parameter values using momentum-based optimizers (Kingma, 2014), thereby reducing mismatch by aligning computations with predicted parameters.

Existing asynchronous pipeline parallelism approaches predominantly adopt a 1F1B schedule, where each device alternates between one forward and one backward passes. To fully eliminate pipeline bubbles, these approaches continuously read minibatches into the pipeline. However, this design causes the severity of parameter mismatch to increase with pipeline depth, ultimately degrading convergence. Parameter stash mitigates mismatch but introduces delayed gradients, while parameter prediction avoids delay but relies on simplified forecasting that often deviates from true parameter values.

To address the aforementioned issues, we propose **A**synchronous **M**ulti-**D**irectional **P**ipeline Parallelism (**AMDP**). AMDP builds upon the bidirectional scheduling concept introduced in Chimera (Li & Hoefler, 2021) for synchronous training, but extends it to a fully asynchronous and multi-directional setting with three components absent in Chimera: 1) a structural mismatch-control mechanism that bounds forward–backward inconsistency to one step; 2) a gradient-accumulation update scheme that confines mismatch effects to minibatches within a window sized by the pipeline depth; and 3) a zero-redundancy optimizer (ZeRO) that efficiently manages the memory overhead of concurrent pipelines. The contributions of AMDP are summarized as follows:

- We propose an asynchronous multi-directional pipeline schedule with a one-step mismatch bound, which limits stage 0 of each pipeline to read only two minibatches before initiating the first backward pass and instantiates multiple concurrent 1F1B pipelines with the optimal number determined analytically based on the pipeline depth.

- We introduce a gradient accumulation update strategy that aggregates gradients across multiple minibatches and applies them in a single update. This reduces communication frequency and ensures that parameter mismatch is bounded to at most one step, affecting only a number of minibatches equal to the pipeline depth.

- We incorporate ZeRO into the multi-directional schedule, ensuring that only one replica of each stage maintains the full optimizer state. This eliminates the redundant memory cost of naive pipeline replication while preserving efficiency.

- Extensive experiments demonstrate that AMDP achieves up to 17% higher throughput compared to

state-of-the-art baselines, with marginal additional memory overhead, significantly accelerating large-scale language model training.

## 2. Related Work

**Synchronous Pipeline Parallelism.** Synchronous pipeline methods launch a new minibatch only after completing the previous update. GPipe (Huang et al., 2019) firstly proposes microbatch-based pipeline training with an "all-forward-then-all-backward" schedule, achieving stable convergence but incurring high activation memory overhead. DAPPLE (Fan et al., 2021) reduces memory cost by adopting a 1F1B schedule that releases activations earlier, and Interleaved 1F1B (Narayanan et al., 2021b) further decreases bubble size via finer stage partitioning. Chimera (Li & Hoefler, 2021) and BitPipe (Wu et al., 2024) improve utilization by running two counter-directed pipelines whose idle periods overlap, and zero-bubble pipeline parallelism (ZB-V) (Qi et al., 2024) theoretically eliminates bubbles by splitting backward computation. Recent weight-passing approaches (e.g. WeiPipe (Lin et al., 2025) and TawPipe (Wu & Chen, 2026)) reduce communication volume by transmitting model weights instead of activations, which can substantially improve bandwidth efficiency in long-context large models training. Despite these advances, synchronous schemes remain fundamentally constrained by sequential dependencies, leading to persistent bubbles and limited throughput.

**Asynchronous Pipeline Parallelism.** Asynchronous methods remove bubbles by updating parameters immediately after each backward pass, at the cost of parameter mismatch. PipeDream (Narayanan et al., 2019) addresses this via parameter stashing, caching forward-pass weights for reuse during backpropagation; PipeDream-2BW (Narayanan et al., 2021a) reduces stash memory by maintaining two parameter versions per device. These approaches avoid mismatch but introduce delayed gradients. Parameter prediction offers an alternative: SpecTrain (Chen et al., 2018) predicts future parameters via momentum states, and XPipe (Guan et al., 2019) improves accuracy using Adam-based prediction. Recent work vNAG (Ajanthan et al., 2025) applies momentum-based extrapolation to alleviate staleness in PipeDream-like pipelines. Prediction-based methods mitigate delay but remain limited by the simplicity of their forecasting rules, which can produce large deviations between predicted and actual weights.

Table 1 summarizes the trade-offs of representative pipeline approaches in terms of bubble ratio, memory overhead, convergence stability, and additional costs in memory, computation, and communication. Synchronous approaches (e.g., DAPPLE, Inter-1F1B, Chimera, and ZB-V) provide stable convergence but incur either high bubble ratios or substantial memory usage. Asynchronous approaches (e.g.,

*Table 1.* Comparison of representative pipeline parallelism approaches. $d$ denotes the pipeline depth, $n$ denotes the number of microbatches per minibatch, and $M_\theta/M_a$ indicate the weight/activation memory footprint of a single stage.

| Approach | Bubble Ratio | Weight Memory | Peak Activation Memory | Extra Mem., Comp., and Comm. | Convergence |
|---|---|---|---|---|---|
| DAPPLE | $(d-1)/(n+d-1)$ | $M_\theta$ | $n \cdot M_a$ | $[\times, \times, \times]$ | Stable |
| Inter-1F1B | $(d-1)/(2n+d-1)$ | $M_\theta$ | $d \cdot M_a$ | $[\checkmark, \times, \checkmark]$ | Stable |
| Chimera | $(d-2)/(2n+d-2)$ | $2M_\theta$ | $d \cdot M_a$ | $[\checkmark, \times, \checkmark]$ | Stable |
| ZB-V | $\approx 0\%$ | $M_\theta$ | $(2d-1) \cdot M_a$ | $[\times, \checkmark, \times]$ | Stable |
| PipeDream | $\approx 0\%$ | $[M_\theta, d \cdot M_\theta]$ | $d \cdot M_a$ | $[\checkmark, \times, \times]$ | Unstable |
| PipeDream-2BW | $\approx 0\%$ | $2M_\theta$ | $d \cdot M_a$ | $[\checkmark, \times, \times]$ | Moderate |
| XPipe | $\approx 0\%$ | $M_\theta$ | $d \cdot M_a$ | $[\checkmark, \checkmark, \times]$ | Moderate |
| vNAG | $\approx 0\%$ | $[M_\theta, d \cdot M_\theta]$ | $d \cdot M_a$ | $[\checkmark, \times, \times]$ | Moderate |
| **AMDP (ours)** | $\approx 0\%$ | $M_\theta$ | $d \cdot M_a$ | $[\checkmark, \times, \checkmark]$ | Near-stable |

PipeDream, PipeDream-2BW, XPipe, and vNAG) eliminate bubbles but suffer from elevated memory requirements and unstable convergence due to parameter mismatch.

**Optimization and Memory Techniques.** Gradient accumulation (Hermans et al., 2017; Li et al., 2020) is widely used in large-scale model training, particularly under pipeline parallelism, where gradients from multiple microbatches are aggregated into a single update. This enables large-batch training with manageable per-device memory, though under synchronous schedules it can exacerbate pipeline bubbles and reduce efficiency. While asynchronous PipeDream-style methods may increase the update interval to slow down parameter changes, this strategy does not impose any structural bound on forward–backward mismatch, which typically still grows with pipeline depth.

Memory-reduction methods such as ZeRO (Rajbhandari et al., 2020; Zhao et al., 2023) partition optimizer states, gradients, and parameters across devices, substantially lowering memory overhead while preserving correctness. AMDP leverages these techniques in an asynchronous, multi-directional setting: gradient accumulation suppresses mismatch within each depth-sized window, and ZeRO eliminates redundant optimizer-state replication across pipelines. Together, these components allow AMDP to achieve both high throughput and near-stable convergence.

## 3. Methodology

As highlighted in Table 1, the limitations of existing pipeline parallelism approaches motivate the design of AMDP, an asynchronous multi-directional pipeline parallelism framework that addresses efficiency, memory usage, and convergence stability. AMDP comprises four interrelated components: (1) a *parameter mismatch control strategy* that structurally limits inconsistencies between forward and backward passes, (2) a *multi-directional scheduling scheme* that maximizes hardware utilization, (3) a *gradient accumulation update strategy* that reduces communication overhead and bounds parameter mismatch, and (4) a *zero-redundancy*

*optimizer* that minimizes redundant memory usage across pipelines.

### 3.1. Parameter Mismatch Control

In asynchronous pipeline parallelism, a minibatch always performs its forward pass at a stage before receiving the corresponding backward pass. As a result, the *parameter mismatch* at a given stage (i.e., the number of parameter updates occurring between a minibatch's forward and backward passes) equals the number of minibatches that have completed forward passes but have not yet started backward passes when the current minibatch enters. In a steady 1F1B schedule, each stage issues a new forward only after completing the previous backward, so the mismatch is exactly "the number of minibatches read before the first backward minus one."

This quantity is determined by two structural limits. First, to maintain bubble-free execution during backward propagation, each stage must execute one additional forward relative to its successor before observing its first backward. Hence stage $i$ can read at most $d - i$ minibatches in this warm-up phase, where $d$ is the pipeline depth. Second, the number of minibatches read at any stage cannot exceed the number read by stage 0, denoted by $n$. Combining both constraints yields the following structural bound:

**Lemma 3.1** (Mismatch Bound). *For any stage* $i \in \{0, \ldots, d-1\}$, *the parameter mismatch satisfies:*

$$\text{mismatch}(i) = \min(n, d-i) - 1. \tag{1}$$

This simple expression explains the instability of existing asynchronous pipelines. When prior work sets $n = d$ to eliminate bubbles, the mismatch becomes $\text{mismatch}(i) = d - i - 1$, which grows linearly with depth and results in increasingly stale gradients.

Figure 2 and Figure 1b illustrate this effect. With $n = 1$, mismatch is zero; with $n = 2$, mismatch becomes one step (e.g., a single update from minibatch 2 occurs between the

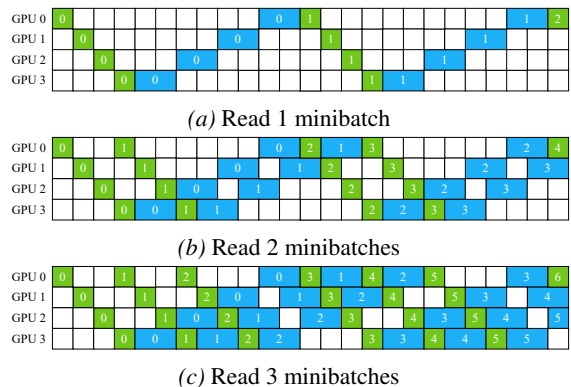

*(a)* Read 1 minibatch

*(b)* Read 2 minibatches

*(c)* Read 3 minibatches

*Figure 2.* Parameter mismatch at stage 0 increases linearly with the number of minibatches read before the first backward pass. The case with the number set to 4 is shown in Figure 1b.

forward and backward of minibatch 3 at stage 0). As $n$ increases, mismatch grows proportionally.

To mitigate mismatch, one would ideally minimize $n$. However, using $n = 1$ requires launching many parallel pipelines to avoid large bubbles, substantially increasing memory usage. Instead, AMDP enforces $n = 2$ by design. From Lemma 1, this guarantees $\mathrm{mismatch}(i) \leq 1$ for all stages $i$, regardless of pipeline depth, multi-node deployment, or multi-directional stage placement (details provided in the Appendix B). A fixed one-step mismatch has important convergence implications:

**Theorem 3.2** (Near-Synchronous Convergence of AMDP). *Let $F$ be an $L$-smooth objective bounded below by $F^\star$. Assume stochastic gradients satisfy $\mathbb{E}[g_t \mid \theta_t] = \nabla F(\theta_t)$ and $\mathbb{E}[\|g_t - \nabla F(\theta_t)\|^2] \leq \sigma^2$. Suppose AMDP enforces a strict one-step mismatch bound $\tau(t) \leq 1$ for all $t$, and choose a stepsize satisfying $\eta L \leq 1$. Then the AMDP iterates satisfy*

$$\frac{1}{T} \sum_{t=0}^{T-1} \mathbb{E}\big[\|\nabla F(\theta_t)\|^2\big] \leq \frac{2(F(\theta_0) - F^\star)}{\eta T} + \eta L \sigma^2 + O(\eta^2),$$

$$(2)$$

*preserving the convergence behavior of synchronous SGD up to a second-order perturbation.*

*Proof.* We follow the standard smooth nonconvex SGD analysis (Bertsekas & Tsitsiklis, 1989; Tsitsiklis, 1994; Lian et al., 2015; Zheng et al., 2017) while incorporating the bounded AMDP mismatch. Since AMDP guarantees $\tau(t) \leq 1$, the stale gradient is evaluated on parameters differing by at most one optimization step. By $L$-smoothness,

$$\|\nabla F(\theta_{t-1}) - \nabla F(\theta_t)\| \leq L\|\theta_{t-1} - \theta_t\| = O(\eta). \quad (3)$$

Define the discrepancy between AMDP and synchronous SGD iterates $\Delta_t = \theta_t - \theta_t^{\mathrm{sync}}$ that evolves according

to $\Delta_{t+1} = \Delta_t - \eta\big(g(\theta_{t-1}) - g(\theta_t^{\mathrm{sync}})\big)$. Combining the bounded delay perturbation with smoothness yields a recursion implying $\mathbb{E}[\|\Delta_t\|^2] = O(\eta^2)$. The Lipschitz continuity of $\nabla F$ then gives

$$\mathbb{E}\big[\|\nabla F(\theta_t) - \nabla F(\theta_t^{\mathrm{sync}})\|^2\big] = O(\eta^2). \quad (4)$$

Combining this perturbation bound with the standard synchronous SGD convergence result completes the proof. An extension to Adam-type optimizers is provided in Appendix B.3. □

### 3.2. Multi-Directional Scheduling

Reducing the number of minibatches read by stage 0 improves convergence but induces significant bubbles, as shown in Figure 2. AMDP avoids this overhead by employing *multi-directional scheduling*: multiple pipelines with complementary directions execute in parallel and fill one another's idle periods. Figure 3 shows a 4-GPU example with two counter-directed pipelines, each GPU hosting two stages. Pipelines process distinct minibatches independently, and GPUs holding the same stage synchronize gradients via all-reduce before applying updates.

AMDP constructs multi-directional schedules in three steps: (i) determining the number of pipelines according to pipeline depth, (ii) assigning their directions while avoiding device conflicts, and (iii) filling residual bubbles. A single bidirectional schedule is insufficient for large $d$, so the number of pipelines must scale inversely with the active ratio $r$—the fraction of time a single pipeline keeps a GPU busy. If stage 0 reads only one minibatch, the pipeline is nearly serialized ($r = 1/d$). With two minibatches, the number of concurrently active GPUs doubles while per-GPU work remains unchanged, giving $r = 2/d$. Since AMDP fixes stage 0 to read two minibatches, setting the number of pipelines to $d/2$ fully eliminates bubbles. Figure 4 shows this scheme for $d = 8$.

Pipeline directions follow the Chimera-style mapping (Li & Hoefler, 2021): for pipeline $j$, if $j$ is even, stage $i$ maps to GPU $(2j + i) \bmod d$; if $j$ is odd, to $(2j - i + d + 1) \bmod d$. For example, in Figure 4, pipeline 0 traverses GPUs $[0, 1, 2, \ldots, 7]$ while pipeline 1 traverses them in reverse order as $[3, 2, 1, 0, 7, 6, 5, 4]$. However, unlike Chimera, AMDP must handle the asymmetry between forward and backward costs, which can create conflicts when multiple pipelines submit work to the same GPU. AMDP resolves these via a simple FIFO rule: the earliest operation proceeds and the later one is deferred (e.g., the conflict between the two backward passes on GPU 2 in Figure 3).

After direction assignment, each block of $d$ minibatches contains three types of bubbles. The middle bubble arises from inherent forward/backward imbalance and cannot be

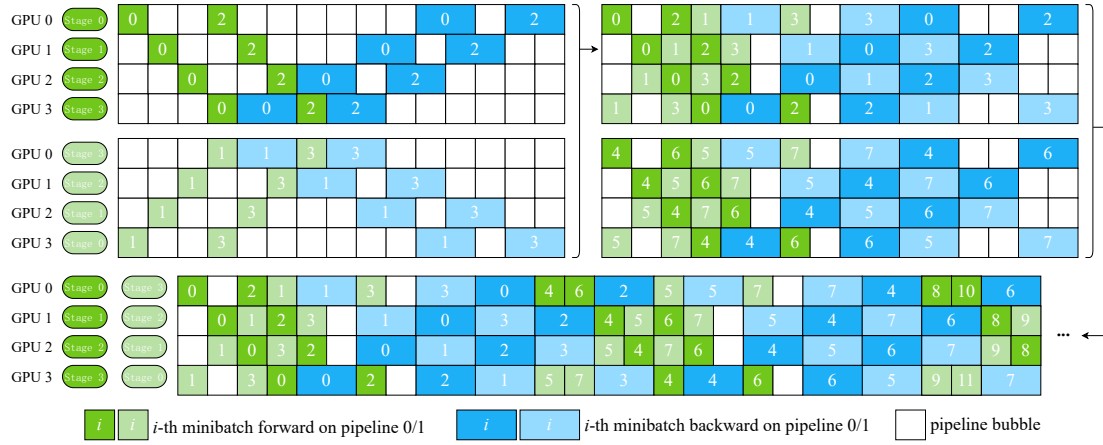

*Figure 3.* Illustration of AMDP bidirectional schedule, with 4 model stages and 4 pipeline devices deployed. Both trailing bubbles from minibatches 0–3 and leading bubbles for minibatches 4–6 are eliminated.

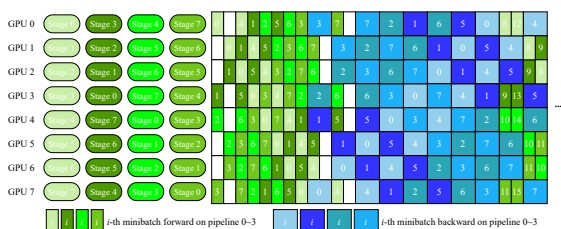

*Figure 4.* Illustration of AMDP multi-directional scheduling for $d$=8. Four (i.e., $d/2$) pipelines with distinct directions is sufficient to filling bubbles.

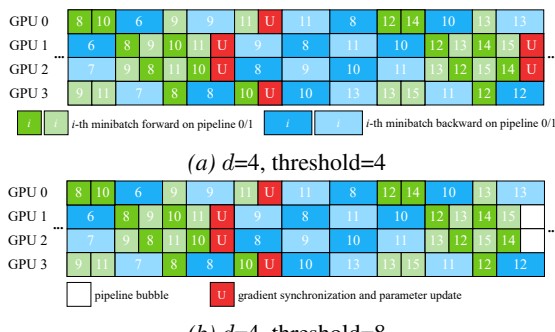

*(a) $d$=4, threshold=4*

*(b) $d$=4, threshold=8*

*Figure 5.* Gradient accumulation update strategy in AMDP. (a) Each minibatch experiences a one-step parameter mismatch. (b) Only the first $d$ minibatches in the window incur mismatch.

removed. It becomes dominant only when the imbalance is extreme or at very large $d$. The leading and trailing bubbles are scheduling gaps and are eliminated via controlled minibatch preloading. At the boundary of each $d$-minibatch segment, AMDP injects additional forward passes to fill idle slots. The number of minibatches inserted equals the floor of the backward-to-forward cost ratio. For instance, in Figure 3 (top-right), preloading minibatches 4 and 6 removes both the trailing bubble of minibatches 0~3 and the leading bubble of minibatches 4~6, yielding balanced and efficient pipeline execution.

### 3.3. Gradient Accumulation Updates

Naively applying the asynchronous multi-directional schedule introduces two drawbacks: (i) an all-reduce after every backward pass, incurring heavy communication overhead, and (ii) disruption of the 1F1B schedule by bubble-filling, which can lead to multi-step mismatches. For example, as shown in Figure 3, two parameter updates (from minibatch 2 and minibatch 4) occur between the forward and backward passes of minibatch 6 on GPU 0.

To address these issues, AMDP introduces a gradient accumulation update strategy. Instead of updating after every

backward pass, gradients are accumulated across multiple minibatches until a predefined threshold is reached. At that point, the accumulated gradients are reduced across devices and applied during the next bubble. This design lowers communication frequency, thereby mitigating communication overhead. Furthermore, it ensures that only the first $d$ minibatches experience a bounded one-step mismatch per update, which enhances training convergence. In practice, the threshold is typically set much larger than $d$, rendering the effect of mismatch negligible. This makes AMDP fundamentally different from PipeDream-style methods, where mismatch typically grows with the number of pipeline stages.

Figure 5 illustrates the gradient accumulation update strategy in AMDP. For clarity, non-essential computation steps are omitted to highlight the effect of accumulation threshold on parameter updates. When the threshold equals the pipeline depth (Figure 5a), each minibatch is subject to a one-step parameter mismatch. With a larger threshold (Figure 5b), only the first $d$ minibatches in each accumulation window are exposed to mismatch, while subsequent

minibatches use consistent parameters. For example, when accumulating gradients from minibatches 8 to 15, parameter updates occur between the forward and backward passes only for minibatches 8˜11.

### 3.4. Zero Redundancy Optimizer

Deploying multiple pipelines increases memory usage, as each device must store parameters, gradients, and optimizer states (e.g., momentum and variance in Adam (Kingma, 2014; Loshchilov, 2017)) for multiple stages. To reduce optimizer-state memory usage without incurring the latency and contention of CPU offloading, AMDP incorporates ZeRO (Rajbhandari et al., 2020). In our design, GPU $i$ is solely responsible for updating parameters of stage $i$, and the optimizer for stage $i$ resides exclusively on GPU $i$. During updates, GPUs holding replicas of stage $i$ send their gradients to GPU $i$ for reduction, replacing the all-reduce used in naive designs; once the update is applied, the new parameters are broadcast back to all replicas. This scheme reduces optimizer state memory on each GPU to $2/d$ of the naive requirement, scaling inversely with the number of pipelines. Furthermore, the communication pattern becomes a reduce followed by a broadcast, which has the same total cost as all-reduce and therefore introduces no additional communication overhead. Synchronization occurs only once per update and does not grow with the number of pipelines.

## 4. Experiments

### 4.1. Experimental Setup

**Hardware and Software.** All experiments are conducted on Linux servers running Ubuntu 20.04 (kernel 5.15) equipped with 8 NVIDIA A800 80GB GPUs interconnected via NVLink 3.0 (400 GB/s), dual-socket 128-core CPUs, and 1 TB DDR4 memory. All methods use the NVIDIA PyTorch container with tag `24.03-py3`[1], ensuring a consistent software environment across baselines. A detailed description of the servers are provided in Appendix A.

**Baselines and Implementation.** We evaluate AMDP against both synchronous and asynchronous baselines: synchronous approaches include DAPPLE (Fan et al., 2021), interleaved 1F1B (Inter-1F1B) (Narayanan et al., 2021b), Chimera (Li & Hoefler, 2021), and zero bubble pipeline parallelism (ZB-V) (Qi et al., 2024); asynchronous approaches include PipeDream (Narayanan et al., 2019), XPipe (Guan et al., 2019), PipeDream-2BW (Narayanan et al., 2021a), and vNAG (Ajanthan et al., 2025). For DAPPLE and Inter-1F1B, we adopt the open source implementation from Megatron-LM (NVIDIA, 2024); for ZB-V and vNAG, we

---

[1] nvcr.io/nvidia/pytorch:24.03-py3

*Table 2.* Configurations of benchmark models.

| Configuration | GPT-style model | BERT-style model |
|---|---|---|
| # Layers | 48 | 32 |
| # Attention Heads | 25 | 32 |
| Hidden Size | 1600 | 1600 |
| Sequence Length | 1024 | 1024 |
| # Parameters | 1557686400 | 1036179458 |

use the codes released by the authors. AMDP, Chimera, and other asynchronous approaches are implemented by us on top of Megatron-LM. The sourse code is released on GitHub[2].

**Models and Datasets.** We test two representative architectures: a GPT-style autoregressive language model and a BERT-style bidirectional encoder, with detailed configurations in Table 2. The GPT-style model is trained on OpenWebText (Peterson et al., 2019), while the BERT-style model is trained on Wikipedia (Devlin et al., 2019).

**Metrics.** We evaluate the performance of each approach using three key metrics: throughput, memory footprint, and training convergence. Throughput, defined as the number of tokens processed per second, reflects the computational efficiency and hardware utilization of each approach. Memory footprint is assessed in terms of both its distribution and peak consumption across devices. The training convergence is assessed by monitoring the training loss against both the number of iterations and the wall-clock time, along with the validation perplexity.

**Training Settings.** Unless otherwise stated, we set the microbatch size to 4 for all approaches. All experiments employ AdamW (Loshchilov, 2017) with mixed-precision training (NVIDIA, 2019). Learning rates and other hyperparameters follow standard configurations commonly used in in GPT/BERT training to ensure fair comparability. Because the author-released implementation of vNAG does not run under our model settings, we adapt AMDP to match their configuration and report the results in Appendix C.

### 4.2. Main Results

**Throughput.** Table 3 summarizes throughput of training the GPT- and BERT-style models on 8 GPUs under varying pipeline depths $d$ and update batch sizes $b$. Key findings are: (1) AMDP consistently achieves the highest throughput across both models and all configurations, outperforming PipeDream-2BW by up to 17%. On GPT-style models, AMDP yields relative gains of 1.01–1.17×, while on BERT-style models, gains range from 1.02–1.13×. These improvements arise because language models exhibit uneven stage

---

[2] https://github.com/Vinsmoke86/AMDP

*Table 3.* Throughput (in ktokens/s) of training the GPT- and BERT-style models on 8 GPUs with NVLink connection. The best and second-best results are **bolded** and underlined. $d$ denotes the pipeline depth and $b$ indicates the number of samples processed per update.

| Model | $d$ | $b$ | DAPPLE | Inter-1F1B | Chimera | ZB-V | PipeDream | XPipe | PipeDream-2BW | AMDP |
|---|---|---|---|---|---|---|---|---|---|---|
| GPT -style | 4 | 16 | 25.2 | 35.4 | 25.3 | 26.3 | 34.5 | 38.5 | 38.6 | **39.1** |
| | 4 | 64 | 36.3 | 39.8 | 30.1 | 30.9 | 34.5 | 40.7 | 41.0 | **42.1** |
| | 8 | 32 | 44.0 | 57.0 | 46.3 | 45.2 | 61.0 | 66.0 | 70.3 | **75.5** |
| | 8 | 128 | 61.9 | 67.5 | 56.0 | 54.4 | 61.0 | 69.7 | 71.6 | **83.7** |
| BERT -style | 4 | 16 | 26.0 | 34.7 | 29.3 | 28.9 | 37.2 | 37.8 | 41.0 | **41.6** |
| | 4 | 64 | 37.1 | 41.0 | 34.5 | 33.8 | 37.2 | 41.7 | 42.9 | **43.6** |
| | 8 | 32 | 45.2 | 37.5 | 52.8 | 40.4 | 65.0 | 73.6 | 74.3 | **78.5** |
| | 8 | 128 | 64.8 | 58.8 | 63.8 | 54.1 | 65.0 | 75.6 | 75.8 | **86.1** |

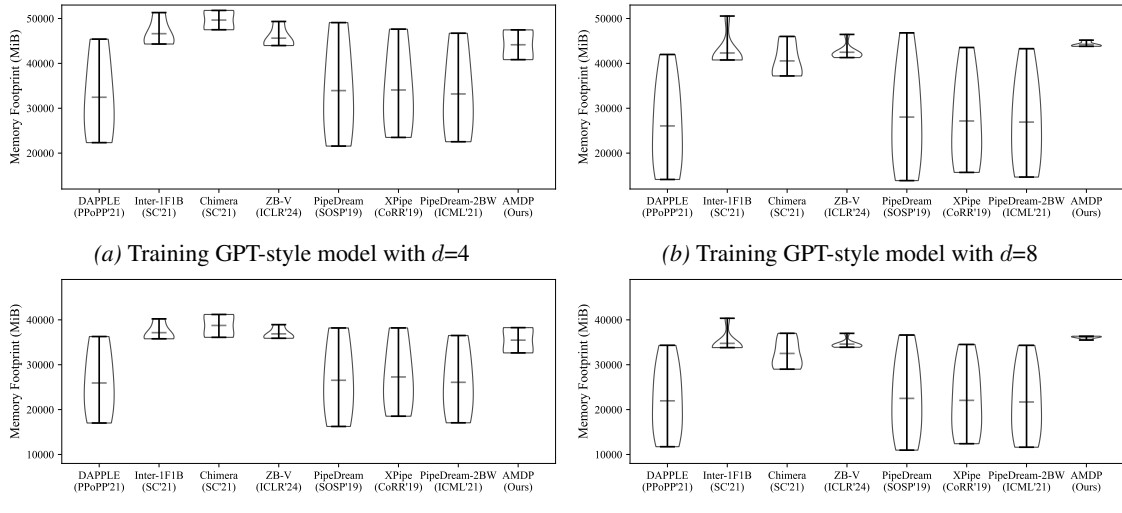

*(a)* Training GPT-style model with $d$=4

*(b)* Training GPT-style model with $d$=8

*(c)* Training BERT-style model with $d$=4

*(d)* Training BERT-style model with $d$=8

*Figure 6.* Memory footprint comparison.

partitioning (e.g., large embeddings in first/last stages), creating imbalance-induced bubbles that AMDP mitigates via multi-directional scheduling, reducing bubble time to $2/d$. (2) The relative advantage of AMDP grows with pipeline depth and update batch size. Larger $d$ exacerbates stage imbalance, while larger $b$ reduces update frequency, both amplifying AMDP's efficiency. (3) Among asynchronous baselines, PipeDream-2BW delivers the strongest throughput, consistently outperforming XPipe and PipeDream. This aligns with its design of double-buffered updates, which eliminate bubbles without additional computation.

Table 4 further reports results on the GPT-style model using a 2-node (16 GPUs) setup, including both pure pipeline parallelism and hybrid pipeline+data parallelism configurations. Key findings are: (1) AMDP continues to outperform all baselines in these larger distributed settings, demonstrating that its benefits extend beyond single-node execution. (2) The throughput advantage of AMDP becomes increasingly apparent on 16 GPUs, potentially because cross-node communication further amplifies the pipeline inefficiencies of baseline approaches.

*Table 4.* Throughput of training GPT-style model on 16 GPUs with NVLink and InfiniBand connections. The best and second-best results are **bolded** and underlined

| $d$ | $b$ | DAPPLE | Inter-1F1B | XPipe | PipeDream-2BW | AMDP |
|---|---|---|---|---|---|---|
| 16 | 128 | 69.4 | 64.5 | 107.4 | 110.1 | **124.3** |
| 8 | 128 | 81.2 | 83.0 | 136.5 | 140.9 | **159.8** |

**Memory Footprint.** Figure 6 illustrates the memory footprint distribution of training the GPT- and BERT-style models on 8 GPUs under two pipeline depths. We summarize the following observations: (1) AMDP achieves more balanced per-GPU memory utilization, which is a direct consequence of its multi-directional scheduling that spreads the workload more evenly across devices. (2) Inter-1F1B exhibits the largest peak memory footprint, as the first GPU must retain additional activations, amounting to 1.5 microbatches for $d = 4$ and 3.5 microbatches for $d = 8$ compared with other approaches. While asynchronous methods generally maintain multiple parameter versions, prior work (Lin et al., 2025) shows that activation storage dominates the memory footprint in modern Transformers. This explains why Inter-

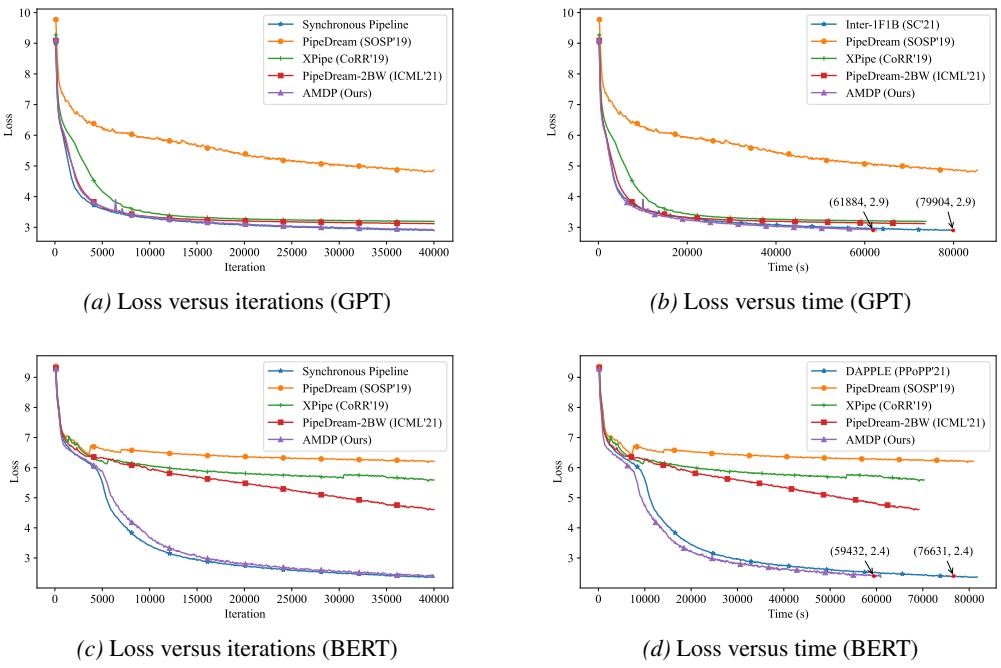

*(a)* Loss versus iterations (GPT)  *(b)* Loss versus time (GPT)

*(c)* Loss versus iterations (BERT)  *(d)* Loss versus time (BERT)

*Figure 7.* Training convergence comparison.

1F1B suffers from significantly higher peak memory. (3) AMDP incurs slightly a higher peak memory footprint than PipeDream-2BW and XPipe due to the need to temporarily retain additional parameters and gradients. However, this overhead is minor relative to the improvements in convergence stability and does not hinder scalability to larger clusters.

**Training Convergence.** Figure 7 presents the convergence results under $d = 8$, $b = 128$, and a learning rate of $1.5 \times 10^{-4}$ over 40k iterations for all methods. Synchronous approaches (e.g., DAPPLE, Inter-1F1B, Chimera, and ZB-V) exhibit nearly identical behavior and are collectively reported as "Synchronous Pipeline". For the loss–time comparisons, we highlight the best synchronous baseline: Inter-1F1B for GPT-style model and DAPPLE for BERT-style model. Importantly, the global batch size $b$ (i.e., samples per update) is kept identical across all methods in each configuration. Consequently, convergence measured in terms of loss versus iterations is equivalent to convergence measured in terms of loss versus processed tokens. This ensures that the comparisons isolate the effect of pipeline scheduling and update semantics, rather than differences in optimization hyperparameters or effective training workload.

Key observations are: (1) AMDP closely tracks the convergence trajectory of synchronous baselines. The small early gap is expected, as the first few iterations accumulate gradients under mild mismatch; once gradients stabilize, AMDP's higher utilization enables it to catch up rapidly. Af-

*Table 5.* Perplexity scores on the validation set at 40k iterations. The best and second-best results are **bolded** and underlined

| Model | PipeDream | XPipe | PipeDream-2BW | DAPPLE | AMDP |
|---|---|---|---|---|---|
| GPT-style | 109.0 | 22.9 | 21.7 | **2.04** | 2.06 |
| BERT-style | 400.1 | 184.8 | 65.7 | 8.3 | **1.7** |

ter 40k iterations, AMDP reaches a training loss of 2.90 on the GPT-style model (vs. 2.88 for Inter-1F1B) and 2.36 on the BERT-style model (matching DAPPLE), confirming the effectiveness of the one-step mismatch bound. (2) AMDP significantly shortens the time to reach target losses: 23% faster than Inter-1F1B on GPT-style model at loss 2.9, and 22% faster than DAPPLE on BERT-style model at loss 2.4. These gains stem from AMDP's higher throughput while maintaining convergence, highlighting its practical value for efficient large-scale models training.

For completeness, Table 5 reports the validation perplexity (PPL) at 40k iterations. AMDP achieves PPL values comparable to those of synchronous baselines at similar training losses, indicating that the bounded asynchrony introduced by AMDP does not cause noticeable degradation in model quality or downstream generalization performance. Together with the loss curves in Figure 7, these results show that AMDP not only preserves near-synchronous optimization behavior during training, but also maintains comparable validation performance after convergence. This observation is consistent with the theoretical analysis in Section 3.1, which shows that AMDP introduces only a second-order

*Table 6.* Throughput of training the GPT-style and BERT-style models on 8 GPUs with different gradient-accumulation thresholds. The best results are **bolded**.

| Model | 1 ($b = 32$) | 2 ($b = 64$) | 4 ($b = 128$) | 8 ($b = 256$) |
|---|---|---|---|---|
| GPT-style | 75.5 | 78.9 | **83.7** | 83.3 |
| BERT-style | 78.5 | 81.0 | **86.1** | 84.6 |

perturbation relative to synchronous optimization.

### 4.3. Gradient-Accumulation Threshold Study

To investigate the impact of the gradient-accumulation threshold in AMDP, we train both GPT-style and BERT-style models on 8 GPUs under varying update batch sizes $b$. Table 6 summarizes the resulting throughput in kilo tokens/s. The results show that a moderate accumulation threshold achieves the best throughput, while further increasing $b$ yields diminishing returns. This behavior reflects the trade-off between update frequency and pipeline utilization. Increasing the accumulation threshold amortizes synchronization and optimizer-update overhead across more minibatches, thereby reducing pipeline bubbles and improving hardware utilization. However, once the pipeline approaches near-full utilization, additional accumulation provides limited further benefit.

The accumulation threshold also affects optimization dynamics. Importantly, AMDP maintains a strict one-step bound on parameter mismatch regardless of the accumulation size, ensuring that gradient staleness remains controlled. As a result, moderate accumulation thresholds are expected to have only limited impact on convergence and may even improve training stability by reducing gradient variance across updates. In contrast, excessively large accumulation behaves similarly to large-batch training, where less frequent parameter updates can slow optimization progress despite improved throughput. This trade-off suggests that a moderate accumulation threshold provides the best balance between training efficiency and convergence behavior.

### 4.4. Ablation Study

To verify the impact of zero redundancy optimizer on memory footprint, we remove it from AMDP and conduct experiment under the same configuration as the training convergence study. Figure 8 shows memory results, from which we observe: without ZeRO, average GPU memory usage rises by 11.4% and 19.4% for GPT-style model (at $d = 4$ and $d = 8$), and by 9.7% and 15.4% for BERT-style model, respectively. Table 7 further indicates that enabling ZeRO improves throughput by approximately 4%, as optimizer state sharding reduces redundant computations. These findings confirm that ZeRO not only alleviates memory pressure but also delivers modest performance gains, making it a

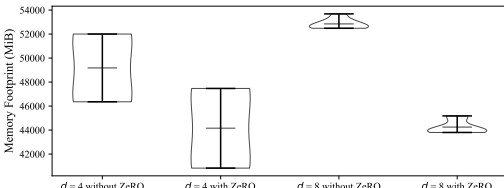

*(a)* Training GPT-style model

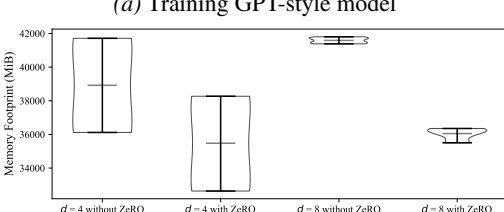

*(b)* Training BERT-style model

*Figure 8.* Memory footprint results of ablation study.

*Table 7.* Throughput results of ablation study. The best results are **bolded**.

| Configuration | GPT-style model | BERT-style model |
|---|---|---|
| without ZeRO | 80.3 (-4.1%) | 82.7 (-3.9%) |
| with ZeRO | **83.7** | **86.1** |

crucial component of AMDP's scalability.

## 5. Conclusions

In this paper, we propose **AMDP**, an **a**synchronous **m**ulti-**d**irectional **p**ipeline-parallel training scheme that achieves state-of-the-art throughput among asynchronous pipeline approaches while maintaining near-synchronous convergence with only marginal additional memory overhead. By structurally bounding forward–backward parameter mismatch and integrating multi-directional scheduling with ZeRO-based state partitioning, AMDP substantially improves hardware utilization without compromising optimization stability. Extensive experiments demonstrate that explicitly limiting pipeline staleness is critical for convergence, while multi-directional execution and ZeRO-based partitioning jointly enable high throughput and low memory footprint. As future work, we plan to explore mismatch-aware learning-rate adaptation technique to further enhance optimization robustness under mild asynchrony.

## Impact Statement

This paper presents work whose goal is to advance the field of Machine Learning. There are many potential societal consequences of our work, none which we feel must be specifically highlighted here.

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

# A. Implementation Details

## A.1. Server and System Configuration

All experiments are conducted on dedicated Linux servers running Ubuntu 20.04 with kernel 5.15. The compute node contains:

- **CPUs:** Dual-socket AMD EPYC 7763 (128 physical cores in two socket).

- **Memory:** 1 TB DDR4-3200, NUMA-balanced across both sockets.

- **GPUs:** 8 NVIDIA A800 80GB SXM5 GPUs, connected via NVLink 3.0 in a hybrid cube-mesh topology, providing 400 GB/s bidirectional bandwidth.

- **Networking:** Local experiments use NCCL's NVLink transport; multi-node configuration supports dual 200 Gbps HDR InfiniBand.

- **Storage:** Parallel NVMe scratch with 7.4 GB/s sequential read bandwidth for fast dataset preprocessing.

The software stack is based on the NVIDIA PyTorch container with tag `24.03-py3`[3], consisting of: PyTorch 2.3.0, CUDA 12.4, cuDNN 9.0.0, NCCL 2.20, Apex, TransformerEngine 1.4.0, and Python 3.10.

## A.2. Code Modifications

We summarize the key extensions made to Megatron-LM to support asynchronous multi-directional execution:

- **Pipeline Engine Extensions.** We extend the Megatron pipeline engine to support asynchronous execution by enabling `async_op` for gradient reductions and parameter updates and introducing an `async_schedules` module. This module implements the scheduling logic for PipeDream, PipeDream-2BW, and AMDP, including forward/backward task queues, causal dependency enforcement, and multi-directional stage assignment.

- **Gradient-Accumulation Mechanism.** AMDP adopts a window-based gradient accumulation mechanism that aligns with ZeRO partitioning. Gradients are accumulated only on the parameter-owner rank, while non-owner ranks store only the transient gradient tensors needed for the reduce operation. This avoids redundant memory usage, guarantees correct accumulation over the update window, and ensures that communication occurs exactly at the intended synchronization boundaries.

- **ZeRO Integration Logic.** We modify the initialization and optimizer-step routines to support multi-directional pipelines under ZeRO. Optimizer states are materialized exclusively on the owner rank of each parameter shard. Each all-reduce is replaced by a *reduce* followed by a *broadcast* of identical communication complexity, triggered once per update window and aligned with AMDP's scheduling.

## A.3. Experiment Workflow

For reproducibility, we outline the common workflow used for all baselines and AMDP:

- **Data Preprocessing.** Training data are first stored as loose JSON (one text record per line) and then converted into Megatron-LM's binary format following the official preprocessing pipeline. All baselines use identical datasets.

- **Pretraining.** Pretraining follows the standard Megatron-LM workflow with additional arguments (e.g., `--enable-fourdirectional-pipeline`). We provide `pretrain_gpt_distributed_async.sh` for AMDP and analogous scripts for baseline methods in the *examples* directory.

- **Logging and Profiling.** Throughput, per-GPU memory, and iteration-time breakdowns are collected using Megatron-LM's profiler with auxiliary custom hooks. Experiments across different depths (4–8) and model families (GPT, BERT) share the same workflow to ensure consistency.

---

[3]nvcr.io/nvidia/pytorch:24.03-py3

# B. Convergence Analysis of AMDP

This part presents a formal convergence analysis of AMDP, demonstrating that it achieves convergence behavior comparable to synchronous adaptive optimization under the bounded delay condition ($\tau_{\max} = 1$). We further prove that this delay bound remains invariant under variations in pipeline depth, multi-node deployment, and multi-directional stage permutations.

## B.1. Problem Setup and Assumptions

Consider the stochastic optimization problem

$$\min_{\theta \in \mathbb{R}^p} F(\theta), \qquad F(\theta) = \mathbb{E}_\xi[f(\theta; \xi)], \tag{5}$$

where $\theta \in \mathbb{R}^p$ denotes the model parameters and $\xi$ is a random minibatch sample.

We adopt the following standard assumptions:

A1. ($L$-**smoothness**) The objective function $F$ is differentiable and satisfies

$$\|\nabla F(x) - \nabla F(y)\| \leq L\|x - y\|, \qquad \forall x, y \in \mathbb{R}^p. \tag{6}$$

A2. (**Unbiased stochastic gradients**) The stochastic gradient estimator $g(\theta; \xi)$ satisfies

$$\mathbb{E}_\xi[g(\theta; \xi)] = \nabla F(\theta), \tag{7}$$
$$\mathbb{E}_\xi\left[\|g(\theta; \xi) - \nabla F(\theta)\|^2\right] \leq \sigma^2. \tag{8}$$

A3. (**Bounded second moment**) There exists $G > 0$ such that

$$\mathbb{E}\left[\|g(\theta; \xi)\|^2\right] \leq G^2, \qquad \forall \theta. \tag{9}$$

A4. (**Bounded adaptive preconditioner**) The adaptive preconditioner $P_t$ is a positive diagonal matrix satisfying

$$0 < c_{\min} \leq \lambda_{\min}(P_t) \leq \lambda_{\max}(P_t) \leq c_{\max}. \tag{10}$$

A5. (**Lipschitz adaptive mappings**) The momentum estimator $m_t = \Phi_t(g_0, g_1, \ldots, g_t)$ and adaptive preconditioner $P_t = \Psi_t(g_0, g_1, \ldots, g_t)$ are Lipschitz with respect to the gradient history:

$$\|m_t - \tilde{m}_t\| \leq L_\Phi \max_{k \leq t} \|g_k - \tilde{g}_k\|, \tag{11}$$
$$\|P_t - \tilde{P}_t\| \leq L_\Psi \max_{k \leq t} \|g_k - \tilde{g}_k\|. \tag{12}$$

**General Adaptive Update Rule.** AMDP employs the generalized adaptive update

$$\theta_{t+1} = \theta_t - \eta P_t m_t, \tag{13}$$

where:

- $m_t$ is a momentum or gradient estimator,

- $P_t$ is a positive diagonal preconditioner,

- $\eta > 0$ is the learning rate.

This formulation includes SGD, Momentum SGD, RMSProp, Adam, AdamW, and AMSGrad as special cases.

For vanilla SGD:

$$m_t = g(\theta_{t-\tau(t)}; \xi_t), \qquad P_t = I. \tag{14}$$

For Adam-type optimizers:

$$m_t = \beta_1 m_{t-1} + (1 - \beta_1)g_t, \tag{15}$$

$$v_t = \beta_2 v_{t-1} + (1 - \beta_2)g_t^2, \tag{16}$$

$$P_t = \text{diag}\left(\frac{1}{\sqrt{v_t} + \epsilon}\right). \tag{17}$$

The logical delay $\tau(t) \in \{0, 1\}$ denotes the number of parameter updates between the forward and backward passes associated with the same minibatch.

For comparison, define the corresponding synchronous adaptive optimizer:

$$\theta_{t+1}^{\text{sync}} = \theta_t^{\text{sync}} - \eta P_t^{\text{sync}} m_t^{\text{sync}}. \tag{18}$$

We further define the trajectory discrepancy:

$$\Delta_t := \theta_t - \theta_t^{\text{sync}}. \tag{19}$$

## B.2. Bounded Parameter Mismatch

We first show that AMDP guarantees a uniformly bounded parameter mismatch.

**Lemma B.1** (Bounded Parameter Mismatch)**.** *Let $d$ denote the pipeline depth and let $n$ denote the number of minibatches that stage 0 may inject before receiving the first backward signal. For any pipeline stage $i \in \{0, \ldots, d-1\}$, the number of parameter updates between the forward and backward passes of the same minibatch at stage $i$ satisfies*

$$\text{mismatch}(i) = \min(n, d - i) - 1. \tag{20}$$

*In AMDP, the scheduler enforces $n = 2$. Consequently,*

$$\text{mismatch}(i) \leq 1, \qquad \forall i. \tag{21}$$

*Proof.* Consider a minibatch $\mathbf{b}$ entering the pipeline. Assume its forward pass reaches stage $i$ at logical time $t$.

Before the backward pass of $\mathbf{b}$ returns to stage $i$, the same stage may process forward passes of subsequent minibatches. The number of such additional forward executions is limited by two independent constraints:

1. (**Pipeline injection limit**) Stage 0 injects at most $n$ minibatches before the first backward signal is received. Therefore, at most $n - 1$ subsequent minibatches may enter the pipeline after $\mathbf{b}$.

2. (**Pipeline propagation limit**) The backward pass of $\mathbf{b}$ must traverse the downstream stages $d - 1, d - 2, \ldots, i + 1$ before reaching stage $i$. During this interval, stage $i$ can process at most one forward pass per downstream stage. Hence, at most $d - i - 1$ additional forward executions are possible.

Combining both constraints yields

$$\text{mismatch}(i) \leq \min(n - 1, d - i - 1) = \min(n, d - i) - 1. \tag{22}$$

Under AMDP scheduling, $n = 2$. Thus,

$$\text{mismatch}(i) = \min(2, d - i) - 1 \leq 1. \tag{23}$$

$\square$

The significance of Lemma B.1 is fundamental: AMDP transforms pipeline parallelism into a bounded-delay stochastic optimization process with $\tau_{\max} = 1$, which is substantially smaller than PipeDream-style training systems whose delay scales with pipeline depth.

## B.3. Near-Synchronous Convergence

We now establish the main convergence theorem.

**Theorem B.2** (Near-Synchronous Convergence of AMDP). *Under Assumptions A1–A5 and the bounded-delay condition* $\tau_{\max} = 1$, *the AMDP iterates generated by* $\theta_{t+1} = \theta_t - \eta P_t m_t$ *satisfy*

$$\frac{1}{T} \sum_{t=0}^{T-1} \mathbb{E}\left[\|\nabla F(\theta_t)\|^2\right] \leq \frac{2(F(\theta_0) - F^\star)}{\eta T} + O(\eta \sigma^2) + O(\eta^2). \tag{24}$$

*where:* $F^\star$ *is a lower bound of* $F$.

*In particular, AMDP preserves the convergence rate of synchronous adaptive optimization up to a second-order perturbation term* $O(\eta^2)$.

*Proof.* Define the discrepancy between AMDP and synchronous trajectories:

$$\Delta_t = \theta_t - \theta_t^{\text{sync}}. \tag{25}$$

Since AMDP guarantees $\tau(t) \leq 1$, the stale gradient is evaluated at parameters differing by at most one optimization step:

$$\|\theta_t - \theta_{t-\tau(t)}\| = \left\| \sum_{k=t-\tau(t)}^{t-1} (\theta_{k+1} - \theta_k) \right\|$$

$$\leq \eta c_{\max} \sum_{k=t-\tau(t)}^{t-1} \|m_k\|. \tag{26}$$

Using Assumptions A3–A4,

$$\mathbb{E}\left[\|\theta_t - \theta_{t-\tau(t)}\|^2\right] = O(\eta^2). \tag{27}$$

By $L$-smoothness,

$$\|\nabla F(\theta_t) - \nabla F(\theta_{t-\tau(t)})\| \leq L\|\theta_t - \theta_{t-\tau(t)}\|$$
$$= O(\eta). \tag{28}$$

From Assumption A5, the perturbations induced in the momentum estimator and adaptive preconditioner satisfy

$$\|m_t - m_t^{\text{sync}}\| = O(\eta), \tag{29}$$
$$\|P_t - P_t^{\text{sync}}\| = O(\eta). \tag{30}$$

Subtracting the synchronous update from the AMDP update yields

$$\Delta_{t+1} = \Delta_t - \eta \left(P_t m_t - P_t^{\text{sync}} m_t^{\text{sync}}\right) \tag{31}$$
$$= \Delta_t - \eta(P_t - P_t^{\text{sync}})m_t - \eta P_t^{\text{sync}}(m_t - m_t^{\text{sync}}). \tag{32}$$

Taking norms and expectations, and using Assumptions A3–A5, there exists a constant $C > 0$ such that

$$\mathbb{E}\left[\|\Delta_{t+1}\|^2\right] \leq (1 + C\eta)\mathbb{E}\left[\|\Delta_t\|^2\right] + C\eta^4. \tag{33}$$

Applying the discrete Grönwall inequality gives

$$\mathbb{E}\left[\|\Delta_t\|^2\right] = O(\eta^2). \tag{34}$$

Again using $L$-smoothness,

$$\mathbb{E}\left[\|\nabla F(\theta_t) - \nabla F(\theta_t^{\mathrm{sync}})\|^2\right] \leq L^2 \mathbb{E}\left[\|\Delta_t\|^2\right]$$
$$= O(\eta^2). \tag{35}$$

Finally, combining this perturbation estimate with the standard convergence guarantee for synchronous adaptive optimization yields

$$\frac{1}{T}\sum_{t=0}^{T-1}\mathbb{E}\left[\|\nabla F(\theta_t)\|^2\right] \leq \frac{2(F(\theta_0) - F^\star)}{\eta T} + O(\eta\sigma^2) + O(\eta^2). \tag{36}$$

This completes the proof. □

**Discussion.** Choosing $\eta = \Theta(T^{-1/2})$ yields the standard nonconvex stochastic optimization rate $O(T^{-1/2})$. Importantly, the additional perturbation introduced by AMDP scales only as $O(\eta^2)$, which is asymptotically dominated by the intrinsic stochastic optimization error. This explains why AMDP empirically exhibits convergence behavior nearly identical to synchronous optimization even when combined with adaptive optimizers such as AdamW.

## B.4. Invariance of the Mismatch Bound

We finally show that the mismatch bound is invariant to deployment topology and scheduling permutation.

**Proposition B.3** (Topology-Independent Mismatch Bound). *Assume that AMDP preserves causal execution order through FIFO scheduling and does not permit logical reordering of updates. Then the mismatch bound in Lemma B.1 depends only on the pipeline injection limit $n$ and the logical stage index $i$. Consequently, when $n = 2$, $\mathrm{mismatch}(i) \leq 1$ remains invariant under:*

- *arbitrary pipeline depth $d$,*

- *arbitrary mapping between logical stages and physical GPUs,*

- *multi-node deployment with bounded communication latency,*

- *arbitrary directional scheduling permutations.*

*Proof.* We formalize execution using a logical event sequence. Each event is represented as $(i, j, \mathrm{type})$, where $i$ is the stage index, $j$ is the minibatch index, and $\mathrm{type} \in \{\mathrm{forward}, \mathrm{backward}\}$. AMDP enforces the following causal constraints:

1. (**Forward-backward dependency**)
$$\mathrm{forward}(i, j) \prec \mathrm{backward}(i, j). \tag{37}$$

2. (**Pipeline forward dependency**) For $i < d - 1$,
$$\mathrm{forward}(i, j) \prec \mathrm{forward}(i + 1, j). \tag{38}$$

3. (**Pipeline backward dependency**) For $i > 0$,
$$\mathrm{backward}(i, j) \prec \mathrm{backward}(i - 1, j). \tag{39}$$

FIFO scheduling further guarantees that the processing order of minibatches is preserved at every stage. Crucially, none of the above constraints depend on physical GPU placement, network topology, communication direction,and multi-node deployment layout.These implementation details may alter wall-clock timing but cannot alter the logical partial order of events. Therefore, the number of forward executions that stage $i$ may perform between $\mathrm{forward}(i, j)$ and $\mathrm{backward}(i, j)$ remains determined solely by:

- the maximum number of injected minibatches $n$,

*Table 8.* Comparison with vNAG. The best and second-best results are **bolded** and underlined. $d$ denotes the pipeline depth and $b$ indicates the number of samples processed per update.

| Model | $d$ | $b$ | Throughput (ktokens/s) | | | Loss (over 50k iterations) | | |
|---|---|---|---|---|---|---|---|---|
| | | | Sync. PP | vNAG | AMDP | Sync. PP | vNAG | AMDP |
| Model-1 | 8 | 8 | 31.0 | 47.1 | **58.5** | **4.05** | 5.12 | 4.09 |
| Model-2 | 8 | 8 | 8.9 | 11.1 | **18.2** | **3.67** | 4.93 | 3.69 |

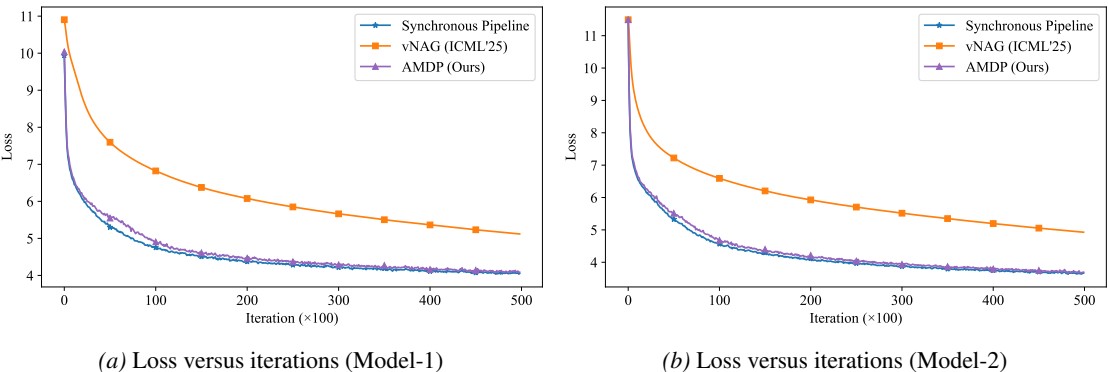

*(a)* Loss versus iterations (Model-1)                    *(b)* Loss versus iterations (Model-2)

*Figure 9.* Training loss.

- the remaining logical pipeline distance $d - i$.

Consequently, $\mathrm{mismatch}(i) = \min(n, d - i) - 1$ is invariant under all deployment and scheduling permutations satisfying the AMDP causal constraints. In particular, when $n = 2$, $\mathrm{mismatch}(i) \leq 1$.

$\square$

## C. Experiments

### C.1. Comparison with recent work

Because the author-released implementation of vNAG (Ajanthan et al., 2025) does not run under our model configurations, we adapt both AMDP and synchronous pipeline approach (i.e., 1F1B implementation in Megatron-LM) to match the experimental setup used in vNAG. The configurations are detailed below:

- Model-1. A GPT-style architecture with a context length of 512, an embedding dimension of 768, 12 attention heads, and 8 transformer layers (approximately 134M parameters), where each layer is treated as one pipeline stage. The minibatch size is set to 8, with a learning rate of 3e-4 and a weight decay of 0.01.

- Model-2. A larger GPT variant that preserves the total number of stages (8) but increases the context length to 1024 and the embedding dimension to 2688, with 24 attention heads (approximately 1B parameters). A learning rate of 1e-4 is applied to all methods for this model.

We train both models from scratch for 50k iterations on the OpenWebText dataset. All experiments are performed on the sever equipped with 8 A800 GPUs. Table 8 reports throughput as well as the final training loss. Figure 9 presents loss trajectories for the experiments. These results highlight two consistent trends: (1) AMDP delivers the highest throughput across both model scales, surpassing vNAG by 24.2% on Model-1 and by 64.0% on Model-2. (2) AMDP tracks the convergence behavior of synchronous baseline much more closely than vNAG, which shows a pronounced deviation. These findings reinforce the central advantage of AMDP, namely its structural one-step mismatch bound, which ensures both efficient utilization and stable optimization dynamics.

