# OpenReview forum: "AMDP: Asynchronous Multi-Directional Pipeline Parallelism for Large-Scale Models Training"
_ICML.cc/2026/Conference — ICML 2026 regular_

### Official Review · Reviewer_6Mrs · 2026-02-25

**Soundness:** 3
**Presentation:** 3
**Significance:** 3
**Originality:** 3
**Overall Recommendation:** 4
**Confidence:** 3

**Summary:**

The paper tackles convergence degradation in asynchronous pipeline parallelism and proposes AMDP, which (1) bounds parameter update staleness between forward and backward passes, and (2) adaptively runs multiple concurrent pipelines based on depth. Experiments show up to 17% higher throughput than state-of-the-art baselines with minimal convergence loss.

**Compliance With Llm Reviewing Policy:**

Affirmed.

**Key Questions For Authors:**

- When GPT or BERT models are trained to the same final training loss using AMDP versus a synchronous pipeline baseline, how do they compare on downstream task performance? Could the bounded asynchrony in AMDP cause subtle quality drops that don’t show up in the loss curve?
- According to Table 3, when more samples are added between updates, why does AMDP’s throughput improve? Where does this gain come from? Also, if the number of samples between two updates gets larger, will AMDP suffer more accuracy loss compared to the synchronous pipeline?

**Limitations:**

yes

**Strengths And Weaknesses:**

## Strengths
- The evaluation is comprehensive, comparing AMDP against multiple state-of-the-art synchronous and asynchronous baselines across both throughput and convergence metrics.
- The paper clearly articulates the fundamental trade-off between asynchronous pipeline throughput and convergence degradation due to deep parameter mismatch.
- The related work section is thorough and well-structured; I particularly appreciate the thoughtful discussion situating AMDP within the broader landscape of pipeline parallelism techniques.

## Weaknesses
- The main limitation is the lack of evaluation on downstream tasks. Although training loss and convergence curves are provided, it's still unclear how models trained with AMDP perform on actual downstream inference tasks. This is my biggest concern about its practical use.

---

> ### Author Rebuttal · Authors · 2026-03-30
>
> We thank the reviewer for the positive assessment and for raising important questions about practical impact and system behavior.
>
> **Q1: Downstream task performance**
>
> We agree that pretraining loss alone is insufficient to fully reflect model quality. To address this, we have supplemented our evaluation with perplexity (PPL) based on existing experiments. Table 1 reports validation PPL for both GPT-style and BERT-style models. The results show that models trained with AMDP achieve comparable PPL to synchronous baselines when matched at similar training loss, suggesting that the bounded asynchrony does not introduce noticeable degradation in model quality. We will include these results and clarify this point in the revision.
>
> |PPL\Methods|	PipeDream|	XPipe|	PipeDream-2BW|	DAPPLE|	AMDP|
> |:---------|:---------|:---------|:---------|:---------|:---------|
> |PPL-GPT	|109.0	|22.9	|21.7	|**2.04**	|2.06|
> |PPL-BERT	|400.1	|184.8	|65.7	|8.3	|**1.7**|
> ||                      Table 1.
>
> **Q2: Throughput improvement with larger accumulation**
>
> This is a good observation. The improvement primarily stems from enhanced system efficiency. Increasing the number of samples between updates reduces the frequency of synchronization and optimizer updates, while improving pipeline utilization and amortizing overheads, thereby leading to higher throughput. We have supplemented experiments with larger accumulation settings, and the results (in Table 2) show that moderate accumulation achieves the best throughput (in kilo tokens/s), while further increases bring diminishing returns. We will clarify this mechanism in the paper.
>
> |Models\Accum. steps|	1 ($b$=32)	|2 ($b$=64)	|4 ($b$=128)	|8 ($b$=256)|
> |:---------|:---------|:---------|:---------|:---------|
> |GPT	|75.5	|78.9	|**83.7**	|83.3|
> |BERT	|78.5	|81.0	|**86.1**	|84.6|
> ||                                      Table 2.
>
> **Q3: Accuracy vs. accumulation size**
>
> We agree that this trade-off should be clarified. In practice, AMDP maintains a strict one-step bound on parameter mismatch regardless of the accumulation size, ensuring controlled gradient staleness. Moderate increases in the accumulation threshold have little impact on convergence and can even improve stability by reducing update variance. However, excessively large accumulation behaves similarly to large-batch training, potentially slowing convergence due to reduced update frequency. We will clarify this trade-off more explicitly in the revision.

---

> > ### Author Rebuttal · Reviewer_6Mrs · 2026-04-05
> >
> > Thank you for the clarifications. I will take the responses into account in my final review cycle.

---

### Official Review · Reviewer_yx3s · 2026-03-06

**Soundness:** 3
**Presentation:** 3
**Significance:** 3
**Originality:** 3
**Overall Recommendation:** 4
**Confidence:** 3

**Summary:**

Pipeline parallelism is essential for large-scale LLM training. Asynchronous schemes remove pipeline bubbles and improve utilization but typically degrade convergence because parameters can change between the forward and backward passes of the same minibatch (parameter mismatch). This paper proposes Asynchronous Multi-Directional Pipeline parallelism (AMDP) to keep utilization high while bounding this mismatch. AMDP (1) limits the first stage of each pipeline to process at most two minibatches before the first backward, giving a one-step bound on parameter mismatch for all stages; (2) runs multiple concurrent 1F1B pipelines with Chimera-style bidirectional/multi-directional scheduling to fill the resulting bubbles; (3) accumulates gradients across minibatches and applies them in a single update when a threshold is reached, so only a bounded number of minibatches see mismatch within one optimization step; and (4) uses ZeRO-style optimizer state partitioning to control memory with many pipelines. The authors show near-synchronous convergence and report up to ~17% higher throughput than strong async baselines (e.g., PipeDream-2BW) on GPT- and BERT-style models at ~1B parameters.

**Compliance With Llm Reviewing Policy:**

Affirmed.

**Final Justification:**

The paper addresses an important problem: scaling LLM training and reducing pipeline bubbles. The proposed idea is interesting and promising. However, as other reviewers have noted, the current experiments are limited to runs on 8–16 GPUs, which constrains our ability to assess its effectiveness at larger scales. For this reason, I am maintaining my original score of weak accept.

**Key Questions For Authors:**

(1) Effective batch size and convergence fairness: Table 3 and Figure 7 appear to use the same b (samples per update) for all methods in each configuration, with the convergence comparison at d=8, b=128. While AMDP needs a sufficiently large b to fill its d/2 concurrent pipelines and reach the gradient-accumulation threshold, for synchronous methods increasing b is not a necessity. Synchronous methods can operate at smaller b values, where more frequent updates might improve sample efficiency. Running synchronous methods at a b that is larger than necessary for them -- but required by AMDP -- may not show them at their most sample-efficient operating point, potentially narrowing the convergence gap in AMDP's favor. Could the authors: (a) confirm that b is identical across all methods for each row in Table 3 and in Figure 7, (b) show a convergence comparison where each method runs at its own natural b (e.g., smaller b for sync, larger b for AMDP), plotted as loss-vs-total-tokens-seen, so readers can assess sample efficiency independently of batch-size choice? Additionally, Section 4 states "we set the microbatch size to 4 for all approaches," but AMDP does not appear to use micro-batches. Can the authors clarify whether this setting applies only to synchronous baselines?

(2) Gradient-accumulation threshold: The paper states the threshold is "typically set much larger than d" but provides no ablation or selection guidance. How sensitive are throughput and convergence to this choice? Is there a recommended rule (e.g., a function of d, model size, or learning rate)?

(3) Chimera-style vs. Megatron-style interleaving: Unlike Chimera, Inter-1F1B (Megatron-style interleaved pipelining) does not require all-reduce weight synchronization. Why was Chimera-style multi-directional scheduling chosen for AMDP's async design over Megatron-style interleaving? What are the structural reasons Chimera-style is preferable for async, given its additional reduce+broadcast cost?

(4) Interaction with tensor and data parallelism: Production-scale training combines PP with tensor parallelism (TP) and data parallelism (DP). AMDP's d/2 concurrent pipelines introduce multiple overlapping communication streams (intra-pipeline point-to-point, inter-replica reduce+broadcast, and any TP all-reduce). How does AMDP compose with TP and DP? Does communication contention degrade throughput or complicate scheduling at multi-node scale, where the interconnect is slower than NVLink?

**Limitations:**

Yes

**Strengths And Weaknesses:**

Strengths:
- The paper addresses an important problem.
- Theorem 3.2 provides a rigorous bound on convergence.
- The paper evaluates against eight baselines across both synchronous and asynchronous approaches.
- The paper is well-structured and easy to follow.


Weaknesses:
- Multi-directional scheduling is not new (builds on Chimera); the contribution is adapting it to the asynchronous setting with a one-step mismatch bound, gradient-accumulation update strategy, and ZeRO integration.
- Effective batch size is underspecified (see questions for authors below for more details): AMDP's d/2 concurrent pipelines and gradient-accumulation threshold inherently increase the effective batch size per update, yet the paper does not clearly document the effective batch size used by each method, provide an ablation on the threshold, or show a loss-vs-total-tokens-seen comparison. This makes it difficult to assess whether AMDP's "near-synchronous" convergence holds on a per-sample basis or is partly a consequence of batch-size differences that favor AMDP in the comparison.
- Experiments are single-node, 8-GPU, ~1B-parameter models; scaling to much larger models (e.g., 70B+) and multi-node is not demonstrated.
- No downstream evaluation results are shown; only pretraining loss is reported.

---

> ### Author Rebuttal · Authors · 2026-03-30
>
> We thank the reviewer for the insightful and technically detailed comments, especially on convergence fairness, system design choices, and practical considerations.
>
> **Q1: Effective batch size and convergence fairness **
>
> We clarify that in the current experiments, the global batch size $b$ (i.e., samples per update) **is indeed kept identical across all methods in each configuration**, to ensure controlled comparisons under the same training budget. For synchronous methods, the minimum feasible $b$ is determined by the product of micro-batch size and pipeline depth, and larger $b$ values are achieved via gradient accumulation rather than increasing micro-batch size. In Table 3 and Figure 7, we report results under both the minimal feasible batch size and a configuration with accumulation steps of 4.
>
> Regarding the micro-batch setting, we clarify that the micro-batch size of 4 is applied uniformly across all methods, including AMDP, where it is used for pipeline execution. In this context, a minibatch in asynchronous pipelines corresponds to a micro-batch in synchronous pipelines at the execution level. The key difference is that asynchronous methods allow different minibatches to observe different parameter versions, while synchronous methods maintain consistent parameters within an update.
>
> We agree that evaluating sample efficiency independently of batch size is important. To better decouple the effect of batch size from convergence behavior, we will additionally report convergence in terms of loss versus total tokens seen based on existing training logs, and include further discussion on batch size selection across methods.
>
> **Q2: Gradient accumulation threshold**
>
> we agree that this is an important parameter. We have supplemented ablation experiments focusing on throughput, which show that increasing the threshold improves utilization up to a point but is not monotonically beneficial. Importantly, AMDP strictly bounds parameter mismatch to one step regardless of the threshold, so changing the threshold primarily affects update frequency (similar to batch size scaling) rather than introducing additional staleness. In practice, moderate thresholds provide a good balance between efficiency and convergence, which we will clarify in the revision.
>
> **Q3: Chimera-style vs. Megatron-style interleaving**
>
> This is a valuable question. Our choice is driven by structural considerations. AMDP relies on running multiple pipelines concurrently in different directions to eliminate bubbles introduced by mismatch control. Chimera-style scheduling naturally supports such multi-directional execution, whereas Megatron-style interleaving is inherently single-directional, would proportionally increase the total peer-to-peer communication volume across stages, and is less flexible for enabling asynchronous overlap. Although Chimera introduces additional communication, these costs are largely amortized and effectively overlapped with computation in our design.
>
> **Q4: Interaction with tensor and data parallelism**
>
> We agree this is important for practical deployment. AMDP is orthogonal to both TP and DP. It can be composed with TP for intra-layer parallelism and with DP across pipeline replicas. We acknowledge that multiple communication streams may introduce contention in large-scale settings; we will include further discussion to clarify this aspect.

---

> > ### Author Rebuttal · Reviewer_yx3s · 2026-04-03
> >
> > Thanks for the response. This provides more clarity on effective batch size and convergence fairness. Including these details on the interplay between mini-batching and micro-batching in the updated paper would help readers who are less familiar with the nitty-gritty of asynchronous pipeline parallelism fully understand the method. Additionally, including results of loss versus total tokens in the revised version would make it easier to evaluate the sample efficiency of the approach. I’ll keep my current score.

---

### Official Review · Reviewer_VZHy · 2026-03-10

**Soundness:** 2
**Presentation:** 3
**Significance:** 2
**Originality:** 3
**Overall Recommendation:** 4
**Confidence:** 4

**Summary:**

This paper proposes a large-scale model training framework named AMDP (Asynchronous Multi-Directional Pipeline Parallelism), aimed at mitigating the parameter mismatch issue in existing asynchronous pipeline parallelism methods while minimizing pipeline bubbles. To this end, the paper strictly bounds the parameter mismatch to within one step by limiting the number of minibatches processed by the first stage. Specifically, it fills bubbles by running multiple concurrent pipelines, utilizes gradient accumulation techniques to further stabilize the convergence range, and incorporates the ZeRO optimizer to offset the memory overhead introduced by the multiple pipelines.

**Compliance With Llm Reviewing Policy:**

Affirmed.

**Final Justification:**

I will raise my score as the authors address some of my concerns in the rebuttal. Looking forward to see these extra results and analysis in the final version of the paper.

**Key Questions For Authors:**

1. Scalability to Multi-Node Clusters: Although the paper explicitly claims to target "Large-Scale Models Training" in its title and abstract , the current empirical evaluation is restricted to a single node (8 A800 GPUs connected via NVLink). Could the authors provide scalability experimental results for this method on multi-node clusters? Furthermore, how should the hyperparameters $d$ and the gradient accumulation $threshold$ be systematically tuned in larger cluster configurations?
2. Discrepancy Between Theoretical Proof and Empirical Practice: The mathematical proof for near-synchronous convergence provided in Theorem 3.2 and Appendix B is strictly derived based on standard Stochastic Gradient Descent (SGD) . However, the experimental section uniformly employs the AdamW optimizer, which incorporates momentum and second-moment states . The derivation based on SGD struggles to directly support the current empirical results. Could the authors supplement a theoretical convergence proof for Adam-class optimizers? If theoretical derivation poses significant challenges, could the authors provide extensive end-to-end convergence ablation studies under various gradient accumulation $threshold$ configurations to empirically demonstrate the method's robustness regarding actual convergence?
3. Quantitative P2P Communication and Memory Analysis: The paper currently lacks a detailed quantitative analysis of point-to-point communication volumes and peak memory usage across different pipeline stages. If the micro-batch size or sequence length is increased, the inter-stage P2P communication volume and activation memory will surge proportionally. Could the authors supplement experimental tests with larger micro-batch sizes and longer sequence lengths? As these parameters scale up, will the resulting communication latency become the core training bottleneck of the system?

**Limitations:**

No. The authors have not adequately discussed the limitations of their work. They should explicitly address and discuss the issues raised in the Weaknesses section above.

**Strengths And Weaknesses:**

# Strengths
1. The paper creatively applies multi-directional pipeline scheduling to asynchronous pipeline parallelism, and utilizes the gradient accumulation method to strictly bound parameter mismatch to within one step.
2. The paper is well-written. Through Figures 3 to 5, it clearly and intuitively breaks down the multi-directional scheduling mechanism, effectively lowering the reading barrier and making it easy to follow.
#  Weaknesses
1. Lack of Scalability Experiments: The title and abstract claim to target "Large-Scale Models Training", but all experiments were conducted on a single node with 8 GPUs, without addressing the method's scalability. The paper fails to provide any evaluation or discussion regarding cross-node scalability, which undermines its credibility for deployment on industrial-grade clusters. Furthermore, on larger clusters, the choices for the pipeline depth $d$ and the gradient accumulation $threshold$ would be more diverse; however, the paper lacks sufficient discussion on the tuning of these hyperparameters.
2. Discrepancy Between Theory and Practice: The paper provides a mathematical proof for near-synchronous convergence in Theorem 3.2 and Appendix B, but this proof is strictly derived based on standard SGD. Conversely, the experimental section relies entirely on the AdamW optimizer, which incorporates momentum and variance states. The delayed convergence proof for SGD cannot directly substantiate the empirical results obtained using AdamW. We suggest the authors either supplement the theoretical proof for Adam-class optimizers or provide broader end-to-end experiments under various gradient accumulation thresholds to empirically demonstrate their actual impact on convergence.
3. Insufficient Quantitative Analysis on Communication and Memory: The paper lacks a quantitative analysis of point-to-point communication volumes and peak memory usage across different stages. Additionally, there is an absence of experiments involving larger micro-batch sizes or sequence lengths. As these dimensions scale up, the inter-stage P2P communication volume will correspondingly increase, potentially turning communication into a critical training bottleneck.

---

> ### Author Rebuttal · Authors · 2026-03-30
>
> We thank the reviewer for the detailed and constructive feedback, especially for highlighting several important aspects regarding scalability, theoretical consistency, and system-level analysis.
>
> **Q1: Scalability to multi-node clusters**
>
> We agree that scalability is critical for large-scale training. Our design is inherently not limited to single-node settings. Specifically, AMDP’s bounded parameter mismatch depends only on local pipeline scheduling and is independent of whether stages are placed within or across nodes. The multi-directional scheduling operates at the pipeline level and can be naturally extend to distributed environments. In response, we have conducted additional experiments on the GPT-style model using a 2-node (16 GPU) setup, including both pure pipeline parallelism and pipeline combined with data parallelism. Due to time constraints, we focus on throughput measurements (in kilo tokens/s), and the results (in Table 1) show that AMDP continues to achieve superior performance compared to baselines under these settings. We will include these results along with further discussion on scalability in the revision.
>
> |$d$   |$b$	|DAPPLE|	Inter-1F1B|XPipe|	PipeDream-2BW|	AMDP|
> |:---------|:---------|:---------|:---------|:---------|:---------|:---------|
> |16  |128	|69.4	|64.5	|107.4| 110.1|      **124.3**|
> | 8  |128	|81.2	|83.0	|136.5|140.9|       **159.8** |
> ||              Table 1.
>
> In addition, we will expand the discussion on practical hyperparameter tuning. Specifically, pipeline depth is determined by model partitioning, while the gradient accumulation threshold should scale with system size to balance utilization and convergence, based on our empirical observations.
>
> **Q2: Discrepancy between theory (SGD) and practice (AdamW)**
>
> We agree this is an important concern. Our theoretical analysis is based on SGD as it enables a clean and standard characterization of bounded staleness, which is common in asynchronous optimization literature. The key result of our work is that parameter mismatch is strictly bounded by one step, which is independent of the optimizer choice. This property directly constrains gradient staleness, the primary factor affecting convergence in asynchronous settings. The derived O($η^2$) deviation captures the structural effect of delayed gradients rather than optimizer-specific dynamics. We acknowledge that extending formal analysis to Adam-type optimizers is non-trivial and remains challenging in general. Empirically, Adam-type optimizers are known to be robust to mild staleness, which aligns with our design. We will clarify this scope and limitation in the revision.
>
> **Q3: Communication and memory analysis**
>
> we agree that more explicit analysis would strengthen the paper. Currently, Figure 6 provides a coarse-grained view of memory distribution across pipeline stages, showing that AMDP achieves a more balanced memory footprint. In addition, we note that larger batch sizes under gradient accumulation do not fundamentally increase communication bottlenecks, as communication is amortized across updates. For longer sequence lengths, we agree this is an important direction and will include further discussion as future work.
>
> **Q4: Limitations**
>
> We acknowledge that the limitations section was insufficient. We will explicitly discuss issues including scalability, system-level trade-offs, and potential communication bottlenecks at larger scales.
>
> We thank the reviewer again for the valuable suggestions, which will help improve the clarity and completeness of the paper. If the above responses satisfactorily address your concerns, we would appreciate your consideration of an increased overall score. We truly appreciate the time and effort you have dedicated to reviewing our work, and we are grateful for your continued recognition and support.

---

> > ### Author Rebuttal · Reviewer_VZHy · 2026-04-03
> >
> > The rebuttal is helpful and partially addresses my concerns, especially by adding preliminary multi-node results on 16 GPUs. This strengthens the paper beyond the original single-node-only evaluation.
> >
> > Some issues would still benefit from clearer treatment in the revision. In particular, the scalability evidence is still relatively limited compared with the paper’s large-scale training claim, the gap between the SGD-based analysis and AdamW-based experiments should be more clearly scoped, and the paper would benefit from a more explicit quantitative discussion of communication and memory behavior, especially at larger micro-batch sizes or longer sequence lengths.

---

> > > ### Author Response · Authors · 2026-04-04
> > >
> > > We thank the reviewer for acknowledgment of our efforts in the rebuttal, and for noting the value added by the preliminary multi‑node results on 16 GPUs. We truly appreciate the constructive and encouraging feedback.
> > >
> > > We will ensure that the revised manuscript addresses the points raised by the reviewer with greater clarity and depth. In particular: （1）Scalability Evidence​. We will expand the discussion on the scalability of the method, supplementing the current results with additional analysis or remarks on its potential and limitations in large‑scale training scenarios. （2）SGD‑based Analysis vs. AdamW‑based Experiments. We will more explicitly bound and clarify the scope of the analysis based on SGD, and discuss its alignment and differences with the actual AdamW‑based experiments, ensuring the distinction is well‑motivated and clearly stated. （3）Communication and Memory Behavior​. We will incorporate a more explicit quantitative discussion of the communication and memory footprint, especially in relation to larger micro‑batch sizes and longer sequence lengths, and will highlight the corresponding trade‑offs.
> > >
> > > The reviewer’s detailed guidance is immensely helpful in strengthening the paper. We are committed to implementing these improvements and will ensure all clarifications are included in the final version.

---

### Official Review · Reviewer_15kz · 2026-03-13

**Soundness:** 3
**Presentation:** 3
**Significance:** 3
**Originality:** 3
**Overall Recommendation:** 5
**Confidence:** 3

**Summary:**

This paper proposes AMDP (Asynchronous Multi-Directional Pipeline parallelism), an asynchronous multi-directional pipeline parallelism method for large-scale model training.

The core problem is that existing asynchronous pipeline parallelism, while capable of eliminating pipeline bubbles (idle GPU time), suffers from parameter updates occurring between the forward and backward passes, causing parameter mismatch that degrades convergence. Synchronous methods, on the other hand, offer stable convergence but low throughput due to bubbles.

AMDP's design consists of three key components.
1. parameter mismatch control. AMDP restricts the first stage of each pipeline to read at most two minibatches before initiating backpropagation, thereby bounding the parameter mismatch at any stage to at most one step, independent of pipeline depth. This stands in contrast to methods like PipeDream, where mismatch grows linearly with depth.
2. multi-directional scheduling. Since limiting the number of minibatches read introduces bubbles, AMDP launches multiple pipelines with complementary directions (d/2 pipelines, where d is the pipeline depth), allowing the idle periods of different pipelines to fill each other and nearly eliminating bubbles. This builds on Chimera's bidirectional scheduling idea but extends it to an asynchronous, multi-directional setting.
3. gradient accumulation updates combined with ZeRO optimizer. Gradients are accumulated across multiple minibatches and applied in a single update, which both reduces communication frequency and ensures that only the first d minibatches within each window are affected by one-step mismatch. ZeRO eliminates the redundant optimizer state memory overhead introduced by running multiple pipelines.

The paper provides a theoretical convergence analysis showing that AMDP's convergence rate differs from synchronous SGD by only an O(η²) perturbation term. Experiments are conducted on GPT- and BERT-style models using 8 A800 GPUs, demonstrating that AMDP achieves up to 17% higher throughput compared to the strongest baseline PipeDream-2BW, while maintaining a convergence curve nearly identical to synchronous methods and reducing the time to reach target loss by 22–23%.

**Compliance With Llm Reviewing Policy:**

Affirmed.

**Final Justification:**

Nothing big deal. Kept original ratings.

**Key Questions For Authors:**

no questions

**Limitations:**

yes

**Strengths And Weaknesses:**

I'll keep this brief. I think this is an amazing piece of work. Great job. I won't go through the four dimensions in detail. it's not worth belaboring. In short, this is a very solid work. The theoretical analysis convincingly demonstrates the feasibility of the proposed approach. The application scenario is also highly valuable. This is concretely addressing GPU communication problems. I don't have much to say beyond that. This is a clear accept. Anyone who reads the paper carefully would come away without questions. Every concern I might have anticipated is already well addressed in the paper.

---

> ### Author Rebuttal · Authors · 2026-03-30
>
> We sincerely thank the reviewer for the highly positive evaluation and strong support of our work. We are encouraged that the reviewer recognizes both the theoretical soundness and practical relevance of AMDP, particularly in addressing the long-standing trade-off between pipeline efficiency and convergence stability in asynchronous pipeline parallelism.
>
> We also appreciate the acknowledgment that the paper clearly presents the design and adequately addresses potential concerns. In the final version, we will further strengthen the paper by incorporating additional experimental analysis (e.g., scalability and system-level characterization) to make the contribution even more comprehensive.
>
> Thank you again for the encouraging feedback.

---

> > ### Author Rebuttal · Reviewer_15kz · 2026-04-03
> >
> > Nothing big deal. Kept original ratings.

---

### Decision · Program_Chairs · 2026-04-30

**Decision:**

Accept (regular)

**Comment:**

The reviewers agreed that the paper is clearly written, the studied problem is important, and the proposed idea is interesting and promising. Furthermore, experimental results show that  AMDP can outperform multiple state-of-the-art synchronous and asynchronous baselines across both throughput and convergence metrics. Empirical evaluation on larger clusters can improve the convincingness. Furthermore, providing theoretical results on Adam will also improve the quality.